# NOMO-1 cells expressing an NF-κB luciferase reporter gene facilitate a simple, rapid monocyte activation test that can detect a wide range of pyrogens

Tomohisa Nanao[1]*, Yuki Marutani[1], Katsuko Sato[1], Tomohiro Mori[1,2], Takeshi Kitagawa[2], Teruaki Oku[3], Takahiro Nishibu[1,2]

1 Bio Science & Engineering Laboratories, FUJIFILM Corporation, Amagasaki-shi, Hyogo, Japan,
2 Biotechnology Center, FUJIFILM Wako Pure Chemical Corporation, Amagasaki-shi, Hyogo, Japan,
3 Department of Microbiology, Hoshi University School of Pharmacy and Pharmaceutical Sciences, Shinagawa-ku, Tokyo, Japan

* tomohisa.nanao@fujifilm.com

## Abstract

Pyrogens, which include endotoxin and non-endotoxin pyrogens (NEPs), act on immune cells in the bloodstream, causing various effects such as fever and endotoxic shock. The limulus amebocyte lysate test, a commonly used endotoxin test in the manufacturing of pharmaceuticals and medical devices, can detect endotoxin but not NEPs. The monocyte activation test (MAT), which uses monocytes, is a testing method included in the European Pharmacopoeia (EP 11.5; 07/2024:20630) that can detect NEPs. The MAT detects the cellular response following activation of Toll-like receptors (TLRs) by pyrogens; released cytokines, such as IL-6, are often the targets of detection. This cytokine release is regulated by the transcription factor NF-κB. In this study, we investigated whether it is possible to detect pyrogens with an NF-κB reporter gene-expressing cell line, using the NOMO-1 cell line as a model monocyte-like line. This study demonstrates that the reporter gene-expressing cells can detect 0.0125 EU/mL lipopolysaccharide (LPS) after 3 hours of incubation, and a stable calibration curve for LPS quantification can be created. Moreover, these cells can detect agonists for TLR1–9 in a concentration-dependent manner. Pharmaceuticals, including blood products and antibody drugs, were used in LPS recovery tests to confirm that they do not interfere with LPS detection. This study demonstrates that NF-κB reporter cells facilitate a simpler, more concise MAT, eliminating the complexity associated with enzyme-linked immunosorbent assays. Moreover, using the NOMO-1 cell line allows for the detection of a wider range of NEPs compared with using existing reporter gene-expressing cell lines.

**Data availability statement:** All relevant data are within the manuscript and its Supporting information files.

**Funding:** This study was supported by FUJIFILM Corporation. The funders had no role in study design, data collection and analysis, or preparation of the manuscript.

**Competing interests:** The authors have declared that no competing interests exist.

## Introduction

The contamination of pharmaceuticals and medical devices with pyrogens can lead to endotoxic shock and inflammatory reactions accompanied by fever and, in the worst cases, even death [1,2]. Therefore, pharmacopeial standards in various countries require the control of pyrogens in the manufacturing of parenteral drugs [3,4]. Pyrogens can be broadly classified into endotoxins and non-endotoxin pyrogens (NEPs) [5,6]. Endotoxins are substances derived from the cell envelope of Gram-negative bacteria, whereas NEPs are derived from the membrane components (e.g., lipoproteins, peptidoglycan, wall teichoic acids, and lipoteichoic acid), DNA, and RNA of Gram-positive bacteria, fungi, and viruses.

The rabbit pyrogen test (RPT), an in vivo pyrogen test that monitors for a temperature increase after injecting samples into rabbits' veins, has been used since the 1940s [7]. However, the RPT has many issues, such as low sensitivity and individual variation [8]. The current pyrogen test, the bacterial endotoxin test using limulus amoebocyte lysate (LAL), has been predominantly used since the 1970s because it is a simple and highly sensitive test that specifically detects endotoxins [9–11]. However, with advancements in endotoxin management techniques and changes in pharmaceutical modalities, more attention is being focused on detecting NEPs.

In light of animal welfare considerations, as of June 2021, the European Pharmacopoeia committed to a 5-year plan to completely replace the RPT [12]. Moreover, the European Pharmacopoeia Commission decided to eliminate the RPT from its monographs during its 179th session in June 2024. As an alternative to the RPT, the monocyte activation test (MAT) using mononuclear cells has been included in the European Pharmacopoeia, with researchers transitioning to its use [13]. This testing method detects the inflammatory response induced by treating human monocytes with pyrogens. The European Pharmacopoeia lists whole blood, peripheral blood mononuclear cells (PBMCs), and monocytic cell lines as sources of cells for the MAT. It provides standards for various parameters, such as the number of donors used and the characteristics of the cells [14,15]. The choice of cells for the MAT continues to be actively debated, with concerns raised about inter-donor variability in PBMC responsiveness, supply stability, and ethical considerations [13].

The fever response is induced when immune cells such as monocytes and macrophages are stimulated, leading to the release of cytokines that transmit information to the hypothalamus [16]. During this process, monocyte-like immune cells engage the Toll-like receptor (TLR) signaling pathways, which are activated by binding to pyrogens. TLR engagement leads to activation of the transcription factor NF-κB and induces the expression of inflammatory cytokines such as TNF-α, IL-6, and IL-1β [17,18].

The purpose of the MAT is to detect substances that induce the release of inflammatory cytokines. MATs therefore frequently rely on the measurement of released inflammatory cytokines using enzyme-linked immunosorbent assays (ELISAs) [19]. However, to ensure sufficient release of inflammatory cytokines, a cultivation time of around 20 hours after sample treatment is necessary, and ELISAs require technical

proficiency and prolonged working hours. Given those limitations, some studies have focused on adopting reporter assays as the detection system for MATs [20–22].

In this study, we focused on NF-κB, a transcription factor that induces the expression of inflammatory cytokines, and constructed an experimental system to measure the activity of an NF-κB luciferase reporter gene as a detection system for the MAT. To select host cells for the reporter gene assay, we investigated commonly available monocyte-like cell lines and macrophage-derived cell lines. Following endotoxin stimulation, IL-6 ELISAs or NF-κB reporter assays were conducted. We identified NOMO-1 cells [23], which were responsive to both lipopolysaccharide (LPS) and NEPs, as suitable host cells and established stable expression of the NF-κB reporter gene in these cells to conduct this study.

## Materials and methods

### Reagents

Agonists for TLR1/2 (Pam3CSK4), TLR2 (HKSA), TLR5 (flagellin), TLR7 (CL-264), and TLR8 (TLR8–506) were purchased from Invivogen (San Diego, CA, USA). The TLR2/6 agonist FSL-1 and TLR4 inhibitor TAK-242 were purchased from Sigma–Aldrich (St. Louis, MO, USA). The TLR9 agonist ODN2395 was purchased from Adipogen (San Diego, CA, USA). Dimethyl sulfoxide (DMSO) was purchased from FUJIFILM Wako Pure Chemical Corporation (Osaka, Japan). The NF-κB inhibitors TPCA-1 and BAY 11–7082 were purchased from Abcam (Cambridge, UK). We also used albumin 25% I.V. 5 g/20 mL (Japan Blood Products Organization, Tokyo, Japan), acyclovir (Sawai Pharmaceutical Corporation, Osaka, Japan), epoetin alfa (Kyowa Kirin, Tokyo, Japan), and romosozumab (AMGEN, Thousand Oaks, CA, USA).

### Cell culture

Human NOMO-1 (JCRB, Osaka, Japan) and NF-κB reporter gene-transfected NOMO-1 were grown in suspension at 37°C in a 5% $CO_2$ atmosphere in RPMI-1640 with L-glutamine and phenol red (FUJIFILM Wako Pure Chemical Corporation) supplemented with 10% (v/v) fetal bovine serum (FBS), 100 μg/mL streptomycin, and 100 U/mL penicillin (FUJIFILM Wako Pure Chemical Corporation). The cells were maintained at a density of $0.5–1.5 \times 10^6$ cells/mL and diluted with fresh medium at 3-day intervals.

### Preparation of stably transfected cell lines

Lentivirus particles were initially created by seeding HEK293T cells (GeneCopoeia, Rockville, MD, USA) at $1.5 \times 10^6$ cells/10 cm dish in 10% FBS-containing RPMI-1640 medium (FUJIFILM Wako Pure Chemical Corporation) 2 days prior to transfection. For transfection, the NF-κB RE vector was introduced using a Lenti-Pac™ HIV Expression Packaging Kit (GeneCopoeia). The cells were then incubated at 37°C for 14 hours and recovery culture was performed in RPMI-1640 medium containing 10% FBS. After 48 hours of incubation, the culture supernatant was centrifuged at 500 *g* for 10 min and then filtered with a 0.45 μm polyethersulfone filter (Merck, Rahway, NJ, USA). Viral titers were measured by quantitative reverse transcription polymerase chain reaction (qRT-PCR) using Lenti-Pac HIV qRT-PCR Titration Kits (GeneCopoeia).

Viral transduction of target cells began with seeding cells in 10% FBS-containing RPMI-1640 medium at $2 \times 10^5$ cells/well in a 24-well plate. The next day, cells were infected at $1 \times 10^9$ copies/well with viral vector solution diluted in medium containing polybrene (Merck), adjusted to a final concentration of 5 μg/mL. Transduction was followed by centrifugation at 1,200 *g* for 60 min at 37°C and incubation in a 5% $CO_2$ incubator for 16 hours. Transduced cells were then cultured in RPMI-1640 medium containing 10% FBS for 2 days to recover, and drug selection was performed with 1 μg/mL puromycin (FUJIFILM Wako Pure Chemical Corporation) for 3 weeks. Finally, the 10 cells were sorted into 96-well plates' each well using a cell sorter, and the clone that was most responsive to LPS was used for all experiments.

## Cell freezing

We added 1α,25-dihydroxyvitamin D3 (Sigma–Aldrich) to $1–10 \times 10^5$ cells/mL in culture medium, and cultured the cells for 3 days. After centrifugation (300 $g$, 5 min, room temperature), the supernatant was removed and the cells were resuspended in CultureSure Freezing Medium (FUJIFILM Wako Pure Chemical Corporation). We aliquoted $5 \times 10^6$ cells/vial and stored them at −80°C.

## Endotoxin standard

United States Pharmacopeia-Reference Standard Endotoxin (RSE) was dissolved in water for injection (Otsuka Pharmaceutical Factory, Tokushima, Japan) at 2,000 EU/mL. Dissolved LPS was diluted to 0.8 EU/mL in culture medium and two-fold dilutions of LPS, with final concentrations ranging from 0.0125 to 0.8 EU/mL, were added to plates.

## MATs

The endotoxin standard and sample solutions prepared with RPMI-1640 with HEPES containing L-glutamine and phenol red (FUJIFILM Wako Pure Chemical Corporation) were added at 50 μL/well to 96-well plates. Each condition was measured in quadruplicate. The frozen cell vials for the MATs were warmed in a 37°C thermostatic water bath for 30–60 seconds and suspended in RPMI-1640 with HEPES containing L-glutamine and phenol red. After centrifugation (300 $g$, 5 min, room temperature), the supernatant was removed and the cells were resuspended in RPMI-1640 with HEPES containing L-glutamine and phenol red with 4% (v/v) FBS; the cells (50 μL/well) were then added to 96-well plates. After incubation (37°C, 5% $CO_2$, 3 hours), Nano-Glo® Luciferase Assay reagent (Promega, Madison, WI, USA) was added to the 96-well plates (100 μL/well). Finally, luciferase activity was determined using a microplate reader (Promega). The signal was expressed in terms of relative light units (RLU). Standard endotoxin was analyzed using a four-parameter logistic curve (according to chapter 2.6.30 of the European Pharmacopoeia, MAT protocol) to create a standard curve for endotoxin pyrogens. To determine sample interference, sample dilutions were analyzed in the presence (Es) or absence (Et) of 0.1, 0.2, and 0.4 EU/mL RSE. The percentage of recovery (R) of the spiked RSE concentration (Sp) was calculated as R = (Es – Et)/Sp × 100 (%). A value of R ranging between 50% and 200% is considered evidence of no interference.

## Cytokine measurement

The IL-6 concentration in the harvested supernatant was determined after 20–24 hours of incubation in a humidified incubator at 37°C in the presence of 5% $CO_2$, using the LBIS Human IL-6 ELISA Kit (FUJIFILM Wako Pure Chemical Corporation).

## qRT-PCR

Total RNA from NOMO-1 cells and PBMCs (PCS-800–011™ from the American Type Culture Collection, Manassas, VA, USA) were isolated using a PureLink RNA Mini kit (Thermo Fisher Scientific, Waltham, MA, USA). cDNAs were synthesized with a SuperScript IV VILO™ Master Mix (Thermo Fisher Scientific). The PCR primers used in this study are shown in S1 Table. Transcript expression was monitored using GeneAce SYBR™ qPCR Mix II (Nippon Gene, Tokyo, Japan), with specific primers. The fluorescence signals were measured in real time with a CFX-96 Real-Time System (Bio-Rad, Hercules, CA, USA). The specificity of the SYBR Green PCR signal was confirmed by melting curve analysis.

## Statistical analysis

The data are expressed as the mean and standard deviation (SD) of the mean. We used Student's t-test for data analysis in this study, and results were considered statistically significant when P values were <0.001.

## Results

### Selection of cells for reporter gene incorporation

To select appropriate cells for incorporation of the reporter gene, we analyzed the expression of TLRs in the monocytic NOMO-1 cell line to assess its suitability for the MAT. We confirmed through qRT-PCR analysis that TLR1 to TLR9 mRNAs were expressed in NOMO-1 cells as well as in PBMCs (Fig 1A). In addition, the mRNA levels of CD14, a molecular marker of monocytes and a known receptor for endotoxin [24], were compared using qRT-PCR. CD14 expression in NOMO-1 cells was higher than that in PBMCs (Fig 1B). These results suggest that NOMO-1 cells may exhibit similar responses to pyrogens as PBMCs.

### Response to LPS

We verified the LPS responsiveness of our NF-κB reporter NOMO-1 cell line. The cells were treated with LPS (ranging from 0.0125 to 0.8 EU/mL) for 3 hours and then luciferase activity was measured. Highly reliable ($R^2 > ;0.99$) four-parameter logistic calibration curves were created on the basis of the luminescence values obtained (Fig 2). The NF-κB reporter cells showed an increase in luminescence values dependent on the concentration of LPS. Treatment with TAK-242 [25], a TLR4 inhibitor, completely inhibited this increase in luminescence values (S1 Fig). Furthermore, a reliable calibration curve ($R^2 > 0.999$) could be created even when treating with concentrations of LPS exceeding 10 EU/mL (S2 Fig).

### Limit of detection and limit of quantification

The limits of detection (LOD) and limits of quantification (LOQ) were calculated using the following expressions:

$$LOD = x + 3s$$

$$LOQ = x + 10s$$

where $x =$ mean value of the 4 replicates for the responses to the blank and $s =$ standard deviation of the 4 replicates of the responses to the blank.

We determined that the LOD were <0.0125 EU/mL and the LOQ were <0.025 EU/mL (Table 1). In many cases, the LOQ of 0.0125 EU/mL was met (Lot 2 and Lot 3); however, there were instances in which the luminescence value exceeded the RLU for 0.0125 EU/mL, as observed in Lot 1. Therefore, the overall LOQ was determined to be 0.025 EU/mL. Additionally, we compared the 0.0125 EU/mL and blank values using Student's t-test, which indicated the difference between the two measurements was statistically significant ($p < 0.001$). Furthermore, in the NOMO-1 cells without the NF-κB reporter gene, IL-6 release was measured using an IL-6 ELISA after treatment with LPS (0.0125 to 0.8 EU/mL) for 20 hours. We observed an LPS concentration-dependent increase in the measured values (S3 Fig). However, the LOD was 0.4 EU/mL, which was higher (indicating lower sensitivity) than the LOD of the reporter assay.

$$x = \text{mean value of the 4 replicates for the responses to the blank}$$

$$s = \text{standard deviation of the 4 replicates of the responses to the blank}$$

$$\textbf{LOD} = \textbf{x} + \textbf{3s} = \textbf{Lot 1} : \textbf{217,100, Lot 2} : \textbf{198,761, Lot 3} : \textbf{169,024} < \textbf{0.0125 EU/mL}$$

$$\textbf{LOQ} = \textbf{x} + \textbf{10s} = \textbf{Lot 1} : \textbf{296,700, Lot 2} : \textbf{264,238, Lot 3} : \textbf{190,896} < \textbf{0.025 EU/mL}$$

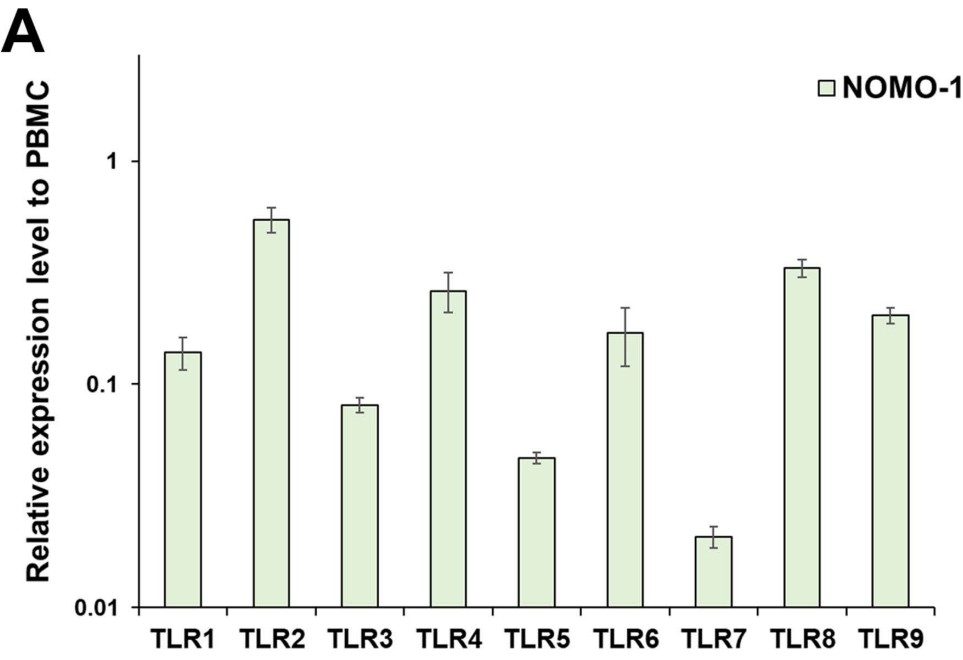

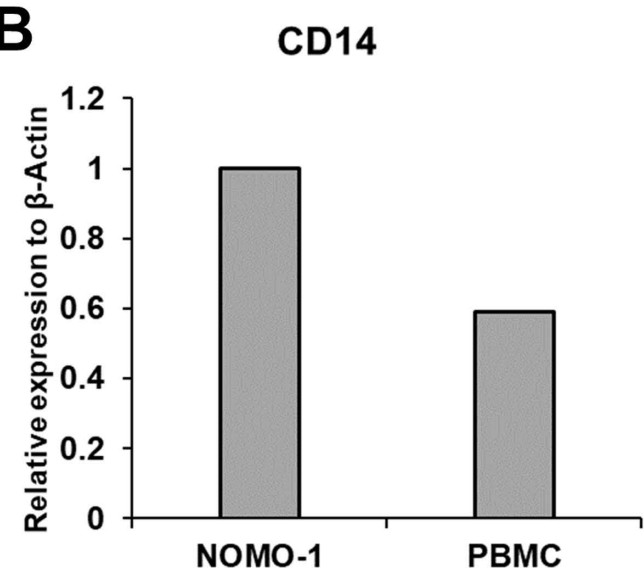

**Fig 1. Comparison of TLR1–9 and CD14 mRNA expression levels in NOMO-1 cells and PBMCs by qRT-PCR.** (A) The relative expression levels of each TLR in NOMO-1 cells relative to the levels in PBMCs; set to 1), normalized to β-actin levels, are shown. RNA was purified from three separate vials, and an independent experiment was conducted for each vial (n = 3). (B) The expression levels of CD14, normalized to β-actin levels, in NOMO-1 cells (set to 1) and PBMCs were compared.

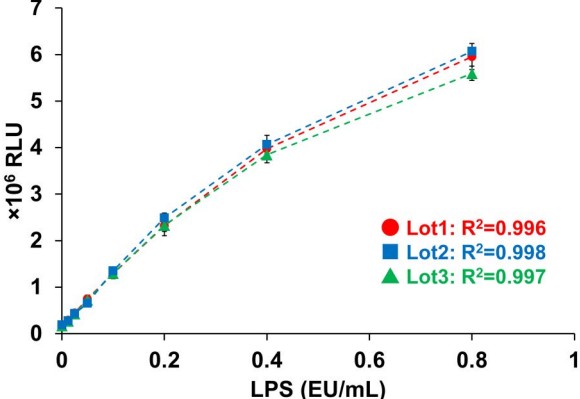

**Fig 2. LPS standard curve.** Four-parameter logistic standard curves were created on the basis of the response of the NF-κB reporter gene to LPS treatment (cells from three separate vials). For each well, a cell suspension of $5 \times 10^4$ cells/50 μL was mixed with 50 μL of LPS standard (0.0125, 0.025, 0.05, 0.1, 0.2, 0.4, and 0.8 EU/mL; the LPS concentration prior to mix) and the plate was incubated for 3 hours. Each LPS concentration was assayed in quadruplicate.

**Table 1. Limits of detections and quantification.**

| LPS(EU/mL) | Average of RLU (4 wells) | | |
|---|---|---|---|
| | Lot 1 | Lot 2 | Lot 3 |
| 0 | 182975 | 170700 | 159650 |
| 0.0125 | 287575*** | 269275*** | 293875*** |
| 0.025 | 443775 | 425250 | 422450 |
| 0.05 | 748550 | 725400 | 741450 |
| 0.1 | 1291000 | 1298500 | 1395000 |
| 0.2 | 2320500 | 2333500 | 2280750 |
| 0.4 | 3976250 | 3847750 | 4221500 |
| 0.8 | 5956000 | 5595000 | 6124750 |
| x+3s | 217074 | 198761 | 169024 |
| x+10s | 296639 | 264238 | 190896 |

***$p < 0.001$, Student's t-test comparing against the blank group.

LOD and LOQ were calculated on the basis of the luminescence values obtained from the reporter assay after LPS treatment.

## Accuracy and precision

The luminescence values measured after treatment with standard LPS were substituted into the calibration curve equation to obtain the concentration of LPS. The accuracy of the obtained LPS concentrations was evaluated for three different frozen lots, and all lots showed high accuracy (80%–120% relative to the standard LPS; Fig 3A). Additionally, the coefficient of variation (CV) values for the calculated LPS concentrations were all < 30% (Fig 3B), indicating high precision.

## Processing time for LPS

In this study, we confirmed that a processing time of 3 hours was sufficient to induce the activation of the NF-κB reporter gene in response to 0.0125 EU/mL LPS. When the LPS treatment period was longer, the luminescence values showed a

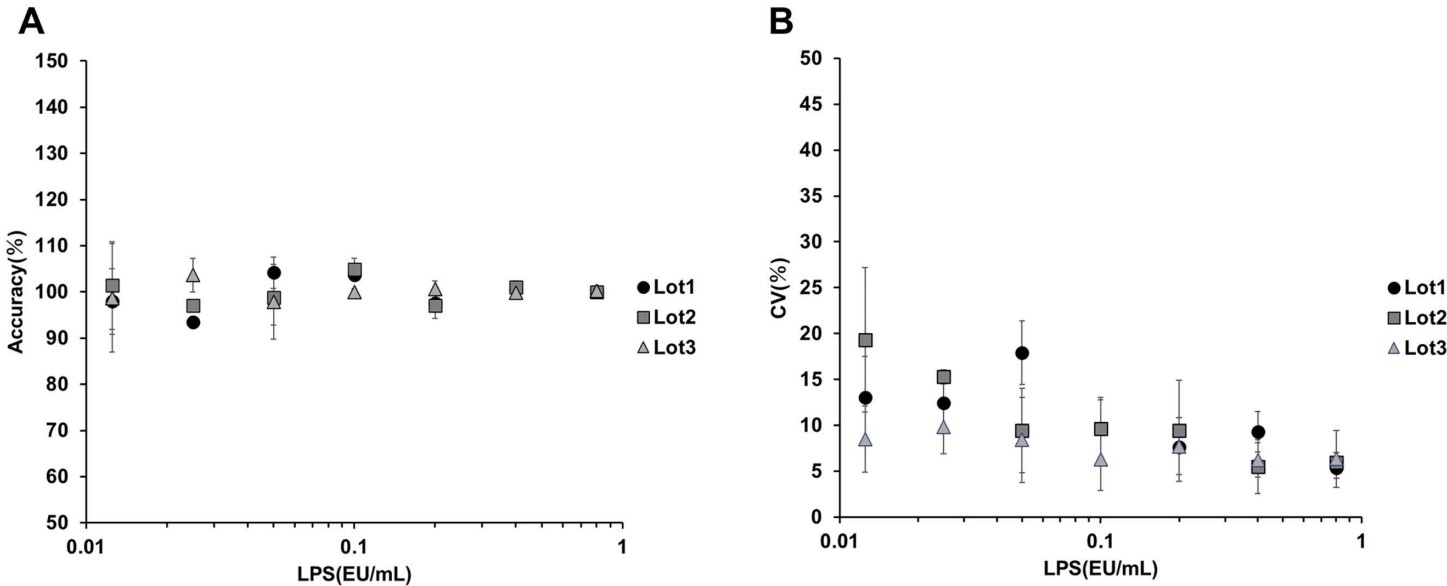

**Fig 3. Accuracy and precision of LPS quantification.** We treated the NF-κB reporter NOMO-1 cells from three freeze lots with standard LPS ranging from 0.0125 to 0.8 EU/mL. A: Accuracy (%) was calculated by reinserting the measured values (RLU) into the four-parameter logistic standard curve and then plotted. Bar (SD): n = 3 vials. B: CV values (%) between vials were determined for the calculated LPS concentrations by reinserting the measured values (RLU) and then plotted. n = 3.

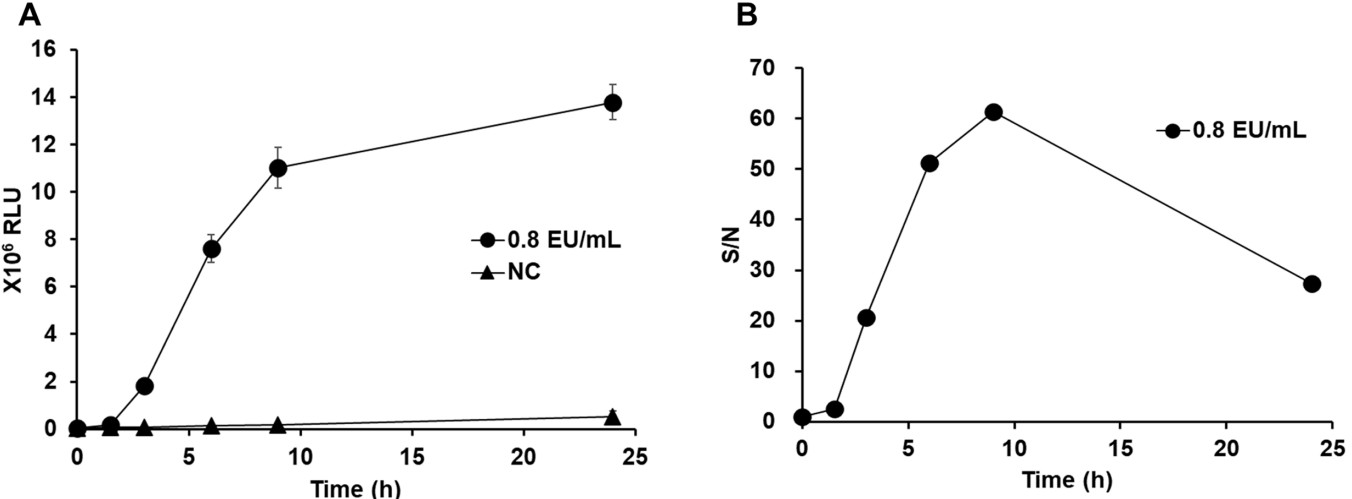

**Fig 4. Changes in response according to incubation time.** The reporter cell line was treated with standard LPS (0.8 EU/mL) and cultured for 0, 1.5, 3, 6, 9, and 24 hours, after which luciferase activity was measured. A: The RLUvalue measured at each time point was plotted. Bar (SD): n = 3 vials. B: The RLU Signal-to-Noise (S/N) value was plotted to compare the samples treated with LPS (0.8 EU/mL) and the untreated samples (NC).

time-dependent increase (Fig 4A). Of note, even under control conditions without LPS treatment, activation of the reporter gene owing to physiological responses was observed, and saturation of the LPS response with prolonged treatment was also observed (Fig 4A). Therefore, the ratio to the background luminescence value decreased, reaching a peak at a processing time of 9 hours (Fig 4B).

## Relationship between NF-κB activation and IL-6 release

Currently, the most widely used detection system for the MAT is the IL-6 ELISA. To confirm the equivalence of NF-κB activation measurement using our reporter assay and IL-6 ELISA as detection systems, we examined the effects of various NF-κB inhibitors on both NF-κB reporter activation and IL-6 release. To investigate whether the activation of NF-κB and the release of IL-6 were coordinated in the reporter cells treated with LPS, we targeted two components of the NF-κB activation signaling pathway (Fig 5A) using two NF-κB inhibitors: TPCA-1 [26], which inhibits IKK, and BAY11−7082 [27], which inhibits the degradation of IκB. Both inhibitors completely inhibited the activation of NF-κB compared with the response in the LPS-treated sample that was not treated with an inhibitor (Fig 5B). An LPS concentration-dependent increase in IL-6 release was observed for the reporter cells. However, the inhibitors completely inhibited IL-6 release, even at low inhibitor concentrations (Fig 5C).

## NEP detectability

To evaluate its ability to detect NEPs, we treated the NF-κB reporter cell line with human TLR1–9 agonists. The resultant luminescence values were substituted into the calibration curve equation, and calculated endotoxin equivalents per milliliter (EE/mL) exceeding the LOD (0.0125 EU/mL) were considered detectable. The NF-κB reporter line showed concentration-dependent luciferase activity in response to all TLR agonists used in this investigation, indicating its ability to detect NEPs

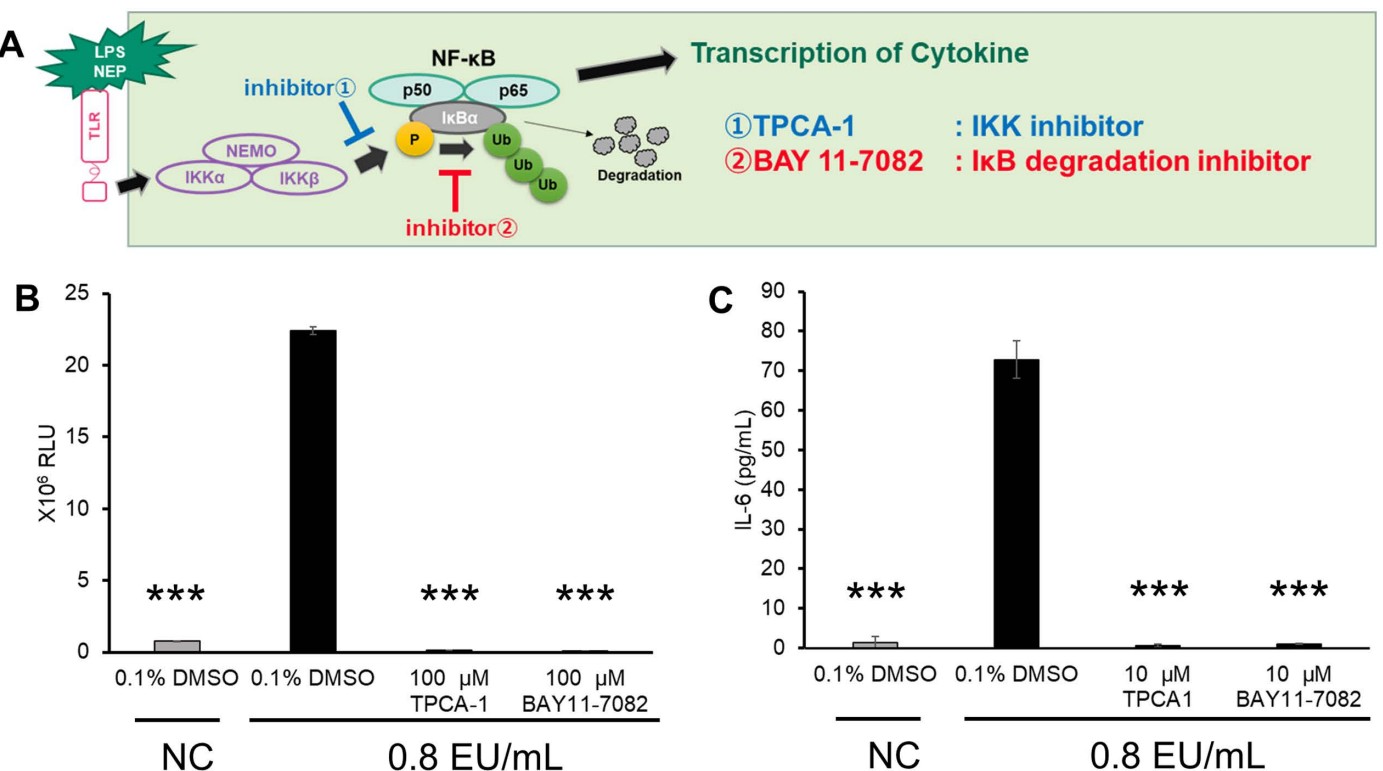

**Fig 5. The effects of NF- κB inhibitors in the NF-κB reporter assay and IL-6 ELISA.** A: Schematic illustration of the targets of two NF-κB inhibitors (① TPCA-1 and ② BAY 11−7082). B: The reporter cells were treated with LPS standard (0.8 EU/mL) and 100 μM inhibitor or 0.1% DMSO; control), and luciferase activity was measured after 3 hours of incubation. C: The reporter cells were treated with LPS standard (0.8 EU/mL) and 10 μM inhibitor or 0.1% DMSO (control), and the culture supernatant was collected after 20 hours of incubation to measure the IL-6 concentration by ELISA. The inhibitor treatment concentration varied according to the incubation time. Bar (SD): n=4, ***: p<0.001.

(Fig 6A–H). Pam3CSK4 (TLR1/2 agonist) was detected at a concentration of 0.63 ng/mL, HKSA (TLR2 agonist) at $5 \times 10^5$ cells/mL, poly(I:C) (TLR3 agonist) at 200 µg/mL, flagellin (TLR5 agonist) at 25 µg/mL, FSL-1 (TRL2/6 agonist) at 5 pg/mL, CL264 (TLR7 agonist) at 5 µg/mL, TLR8–506 (TLR8 agonist) at 0.1 µg/mL, and ODN2395 (TLR9 agonist) at 50 µg/mL.

## Recovery tests with pharmaceuticals

LPS recovery tests were conducted in the presence of pharmaceuticals, including blood products and antibody drugs. LPS was spiked into pharmaceutical dilutions and the recovery rate was determined. If the recovery rate was within the range of 50%–200%, the pharmaceutical agent was considered not to have interfered with the assay.

Blood products known to interfere with the LAL test, such as 25% human serum albumin and human erythropoietin, showed a recovery rate with an error of less than ±15% at dilution factors below the maximum valid dilution (MVD) (Table 2). The presence of acyclovir, which exhibits high alkalinity (pH 10.7–11.7), resulted in a recovery rate very close to the criterion (50%) at a high concentration (10-fold dilution), and exhibited sufficient recovery (over 80%) at dilution factors within the MVD (100- and 400-fold dilutions) (Table 2). We also investigated the antibody drug romosozumab. Even at a high concentration (100-fold dilution), which was significantly higher than the MVD (>10,000-fold dilution), we observed a recovery rate close to 100% after spiking with romosozumab (Table 2). These results indicate that the

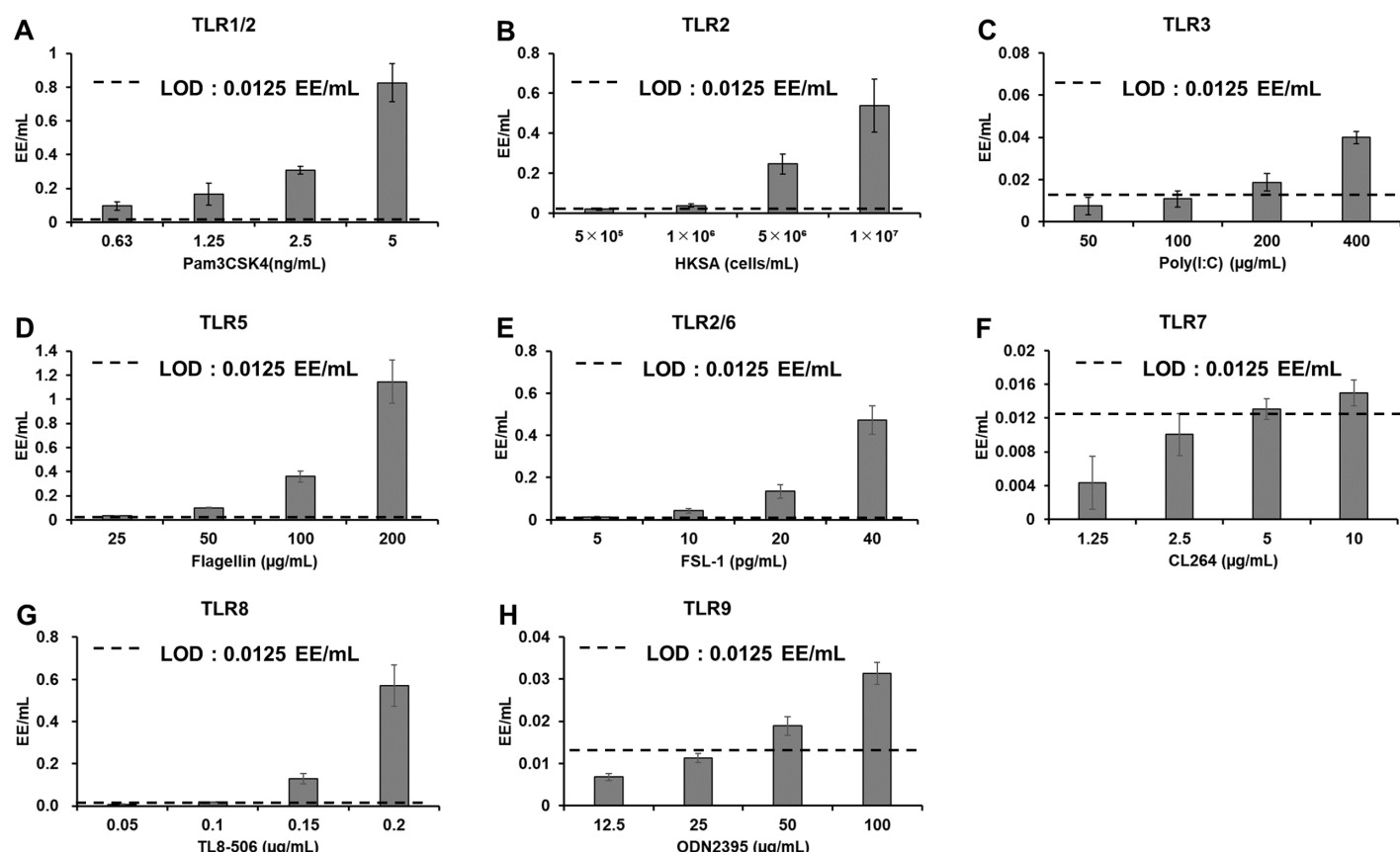

**Fig 6. Detection of NEPs.** (A–H) The NF-κB reporter cells were treated with agonists for TLR 1/2, 2, 3, 5, 2/6, 7, 8, and 9, and luciferase activity was measured after 3 hours of incubation. The measured RLU values were substituted into the four-parameter logistic standard curve obtained from treatment with the LPS standard to calculate the endotoxin equivalents per milliliter (EE/mL). If the calculated value exceeded the LOD of 0.0125 EU/mL, the agonist was considered detectable. Bar (SD): n = 3 vials.

NF-κB reporter cell assay allows for appropriate recovery of LPS in the presence of pharmaceuticals with different characteristics.

## Discussion

In this study, we stably transduced an NF-κB reporter gene into the NOMO-1 cell line, a pre-monocytic cell line, and verified the ability of the reporter cells to detect pyrogens and quantify endotoxins.

### Expression of TLRs and CD14 in NOMO-1 cells

During the host cell selection process, we confirmed the expression of the most important receptors binding to pyrogens, TLRs, in NOMO-1 cells by qRT-PCR. NOMO-1 cells expressed TLR1–9 in a similar pattern to PBMCs (Fig 1A). Although some TLRs (TLR5 and TLR7) showed lower expression levels than in PBMCs, they were not entirely absent, and responses to their corresponding agonists were confirmed in NOMO-1 cells (Fig 6). Furthermore, the expression level of CD14, a monocyte marker [24], was confirmed to be equivalent to or higher than in PBMCs in NOMO-1 cells (Fig 1B). These results suggest that NOMO-1 cells possess monocyte-like characteristics and likely have a similar capacity to respond to pyrogens compared with PBMCs.

### Response to LPS and creation of a standard curve

ELISAs are commonly used as the detection system for MAT assays [14,15]. However, the luminescence measurement used for the luciferase reporter assay has a wider dynamic range than the absorbance measurement used for ELISA, enabling the detection of changes in signal response with greater sensitivity. Treatment of the NF-κB reporter NOMO-1 cells with LPS resulted in a concentration-dependent increase in luminescence values, allowing for the creation of a reliable ($R^2 > 0.99$) LPS standard curve (Fig 2). The LOD for endotoxin in the NF-κB reporter assay was 0.0125 EU/mL, more than 30 times lower (indicating higher sensitivity) than the LOD for the IL-6 ELISA (LOD = 0.4 EU/mL) using the NOMO-1 parental cell line (S3 Fig, Table 1). Furthermore, the MAT using the NF-κB reporter NOMO-1 cells allowed for a shorter incubation time (3 hours) than required for ELISA (20 hours) by detecting the activation of a transcription factor, which

**Table 2. Recovery test for pharmaceuticals.**

| Drug | MVD | Fold-dilution | Spike recovery (%) n = 3 | | | Interference |
| --- | --- | --- | --- | --- | --- | --- |
| | | | 0.1 EU/mL | 0.4 EU/mL | 0.8 EU/mL | |
| Albumin 25%I.V. 5g/20 mL | 480 | 10 | 121.6 | 98.9 | 97.5 | N |
| | | 100 | 134.8 | 119.4 | 110.3 | N |
| | | 400 | 109.5 | 110.3 | 108.0 | N |
| Aciclovir 25 mg/mL | 500 | 10 | 67.5 | 63.7 | 50.2 | N |
| | | 100 | 89.7 | 87.8 | 82.3 | N |
| | | 400 | 96.0 | 95.6 | 99.2 | N |
| Epoetin Alfa 750 I.U./0.5 mL | 12000 | 100 | 99.3 | 103.1 | 105.2 | N |
| | | 500 | 92.9 | 100.5 | 102.6 | N |
| | | 1000 | 87.9 | 98.0 | 98.9 | N |
| Romosozumab 105 mg/1.17 mL | 10200 | 100 | 109.9 | 110.5 | 112.6 | N |
| | | 500 | 94.9 | 94.3 | 98.6 | N |
| | | 1000 | 95.0 | 92.0 | 92.3 | N |

A reporter assay was conducted for samples spiked with pharmaceuticals diluted to three concentrations below the MVD and spiked with 0.1, 0.4, and 0.8 EU/mL LPS. The recovery rates for the LPS concentrations were determined on the basis of the luminescence values obtained for the spiked standard LPS. The spike recovery rate was the average of the results obtained from three tests (n = 3 vials). N: No interference

serves as a precursor to cytokine release. Moreover, the post-reaction measurement can be performed simply by adding the substrate for luciferase, avoiding cumbersome ELISA procedures and reducing variability between experiments. Therefore, we expect the use of the NF-κB reporter NOMO-1 cells for the MAT to greatly contribute to the implementation of rapid, convenient, and highly reproducible tests.

### Accuracy and precision of the standard curve

Regarding the LPS standard curve, good accuracy and precision were observed (80% < accuracy < 120%, CV < 30%) across multiple freeze lots (Fig 3A, B). Additionally, the LOD was calculated to be 0.0125 EU/mL and the LOQ was determined to be 0.025 EU/mL (Table 1), indicating high sensitivity to endotoxin. The sensitivity of our system is comparable to those of systems in previous reports on MATs using PBMCs [28]. In accordance with these results, the presence or absence of NEPs can be determined on the basis of whether the luminescence values exceed the LOD using the standard curve. Notably, in ELISAs using PBMCs, the measurement values often reach a plateau around 1 EU/mL owing to the absorbance dynamic range. However, with luminescence detection, a highly accurate standard curve could be created even when processing endotoxin concentrations exceeding 10 EU/mL (S2 Fig). Furthermore, we found that beyond 9 hours, the S/N decreased owing to a rise in background activity and the attenuated increase in luminescence (Fig 4). This indicates that prolonged cultivation exceeding 9 hours is not necessary even for measurements requiring higher sensitivity.

### Relationship between the NF-κB reporter assay and IL-6 ELISA

Currently, the main kits available for performing the MAT are ELISAs to detect cytokines (such as IL-6 and TNF-α) released by cells in the culture supernatant upon stimulation with pyrogens. The release of IL-6 from the NF-κB reporter cells in response to endotoxin stimulation was almost completely abolished when NF-κB inhibitors (IKK inhibitors and NF-κB phosphorylation inhibitors) were used, suggesting strong dependence of the NOMO-1 cells' response to LPS on the activation of NF-κB (Fig 5A–C). On the basis of these results, we believe a MAT using the NF-κB reporter assay can detect responses equivalent to IL-6 ELISA-based tests.

### Detection of NEPs

The NF-κB reporter NOMO-1 cell line is capable of detecting agonists corresponding to all known human TLRs, for which ligands are known, with a sensitivity of 0.0125 EE/mL (LOD) or higher (Fig 6). Although previous reports have focused on MATs using NF-κB reporter genes, they have not presented data demonstrating a response to all TLR1–9 agonists, exceeding previous studies on detectable types of Pyrogens [20–22]. Our NF-κB reporter NOMO-1 cell line detected agonists for TLRs 1–9, but the agonists for TLR3, 7, and 9, which localize to endosomal membranes, elicited lower luminescence values compared with agonists for the other TLRs. Similar trends in the lower sensitivity to these TLRs have been found in previous reports using PBMCs for IL-6 ELISAs [19]. One possible explanation for the low sensitivity to these TLRs is that monocytes, which are primarily used in the MAT, may not actively propagate signals for TLR3, 7, and 9.

### Recovery in the presence of pharmaceuticals

In addition to using blood products (25% albumin) that have been reported to interfere with the LAL test, we used various pharmaceuticals, including antibody drugs, to perform LPS recovery tests. All the pharmaceuticals used in this study showed recovery rates ranging between 50% and 200% at dilution factors below the MVD for all concentrations of LPS used in the standard curve (Table 2). These results suggest that the MAT method using our NF-κB reporter NOMO-1 cell line is applicable for a wide range of pharmaceutical testing. Furthermore, it would be worthwhile to consider the feasibility of using these cells to perform pyrogen tests on medical devices in the future.

## Conclusions

In this study, we demonstrated the feasibility of conducting MATs using NF-κB reporter cells. We confirmed that the induction of NF-κB expression and the release of IL-6, both induced by exposure to pyrogens, are likely to occur within the same signaling pathway. The NF-κB reporter cells will allow for a stable supply of cells given their clonality. Furthermore, the simplicity of the reporter assay should enable more reproducible testing, reducing the burden on the quality control process by providing rapid, reliable measurements. Therefore, the NF-κB reporter cells hold promise as an alternative detection modality for MATs, which currently mainly rely on the use of PBMCs and IL-6 ELISAs.

## Supporting information

**S1 Fig. Lipopolysaccharide activates NF-κB through Toll-like receptor 4 in reporter cells.** TAK-242, a Toll-like receptor (TLR) 4 inhibitor, was added (final concentration, 1 μM) to the mixture of lipopolysaccharide (LPS) standard and the stable reporter cells and incubated for 3 hours. Bar (SD): n = 4 wells.
(TIF)

**S2 Fig. NF-κB reporter cells exhibit a wide dynamic range of responses to lipopolysaccharide.** A four-parameter logistic standard curve was created on the basis of the response of the NF-κB reporter gene to lipopolysaccharide (LPS) treatment. For each well, a cell suspension of $5 \times 10^4$ cells/50 μL was mixed with 50 μL of LPS standard (0.0125, 0.025, 0.05, 0.1, 0.2, 0.4, 0.8, 2.5, 5, 10, 25, 50, and 100 EU/mL) and incubated for 3 hours. Each LPS concentration was measured in quadruplicate.
(TIF)

**S3 Fig. IL-6 release by NOMO-1 cells upon lipopolysaccharide treatment.** A cell suspension of $1 \times 10^5$ cells/100 μL was mixed with 100 μL of lipopolysaccharide (LPS) standard (0.0125, 0.025, 0.05, 0.1, 0.2, 0.4, and 0.8 EU/mL) in each well and incubated for 22 hours. After incubation, 50 μL of the culture supernatant was used for an enzyme-linked immunosorbent assay. Each LPS concentration was measured in quadruplicate.
(TIF)

**S1 Table. Primers used for quantitative reverse-transcription polymerase chain reaction.**
(TIF)

## Author contributions

**Conceptualization:** Teruaki Oku.

**Investigation:** Tomohisa Nanao, Yuki Marutani, Teruaki Oku.

**Project administration:** Takahiro Nishibu.

**Writing – original draft:** Tomohisa Nanao.

**Writing – review & editing:** Tomohisa Nanao, Yuki Marutani, Katsuko Sato, Tomohiro Mori, Takeshi Kitagawa, Teruaki Oku, Takahiro Nishibu.

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
