## [Decision Letter · Decision Letter 0]

PONE-D-24-22098NOMO-1 combination with the Luciferase reporter gene offers a simple and rapid Monocyte Activation Test that can detect a wide range of pyrogen.PLOS ONE

Dear Dr. Nanao,

Thank you for submitting your manuscript to PLOS ONE. After careful consideration, we feel that it has merit but does not fully meet PLOS ONE’s publication criteria as it currently stands. Therefore, we invite you to submit a revised version of the manuscript that addresses the points raised during the review process.

We have received one thorough and insightful review of your work. After careful consideration, I have determined that this single review provides sufficient basis for a decision. The reviewer's comments are comprehensive and align well with our publication criteria. Please note that this approach is not our standard practice. However, given the circumstances and the substantive nature of the review received, we believe this decision serves both the authors and the journal well.

We look forward to receiving your revised manuscript.

Kind regards,

Rui Tada, Ph.D.

Academic Editor

PLOS ONE

 [T.O. was supported by FUJIFILM Corporation, Japan.].  

Reviewers' comments:

Reviewer's Responses to Questions

**Comments to the Author**

1. Is the manuscript technically sound, and do the data support the conclusions?

Reviewer #1: Partly

2. Has the statistical analysis been performed appropriately and rigorously? 

Reviewer #1: Yes

3. Have the authors made all data underlying the findings in their manuscript fully available?

Reviewer #1: Yes

4. Is the manuscript presented in an intelligible fashion and written in standard English?

Reviewer #1: No

5. Review Comments to the Author

Reviewer #1: In the presented work, Nanao and colleagues demonstrated the establishment of a NF-kB gene reporter assay employing the NOMO-1 monocyte like cell line as a suitable model for pyrogen detection. In their work, the authors showcased the capacity of the established reporter cell line to detect both enodotoxin and non-endotoxin based pyrogens at relatively short incubation period (3 h) and low LOD, relying on the transcription factor NF-kB activation. Hence, the illustrated model provides an attractive tool to further advance the monocyte activation test replacing the tedious and time intensive ELISA based MAT with a more sensitive and automated readout.

Overall Comment: The authors have demonstrated a tool to further advance the field of pyrogenicity testing, nevertheless the novelty of the presented work is majorly lacking due to the abundance of similar and more advanced models already published with gene reporter assays employing more relevant immune cells (e.g. more primary like cells, iPSC- derived reporter macrophages). Moreover, the presentation of the work requires drastic revision with regard to language and profound/ critical result description and discussion. We therefore have strong concern to publish the work in the PLOS ONE journal due to lack of sufficient merit and novelty for the presented work.

Below the authors may find helpful comments to improve the overall quality of their work,

Comment 1 Given the fact that the introduced NOMO-1 cell line is not yet approved by respective regulators or introduced to the MAT context, the authors are strongly advised to start with detailed phenotyping of the used cell line with respect to monocyte/macrophage markers, maturity, and functional analysis in comparison to proper standard primary monocytes. This should be the first critical analysis that can determine further the suitability of the line for the MAT assay.

Comment 2 While the assessment of TLR expression is a critical factor to allow for the cell line inclusion in the MAT context, nevertheless the performed and demonstrated comparison of the NOMO-1 TLR expression profile with respect to MM6 or THP1 cells is not a useful or an informative comparison. As the illustrated reference lines (MM6/THP1) are highly debatable and doubted for their suitability in the MAT, with the fact that THP1 cells already failed the MAT validation studies, and MM6 validation status is incomplete. Hence, the TLR expression analysis needs to be compared to and evaluated against primary monocytes expression. In this line, a functional comparison to established MAT cell lines will add more reliable data than comparing only the expression levels.

Comment 3 The authors need to be more accurate in their introduction concerning properly defining NEPs, and the approved cell sources for MAT that also included whole blood MAT.

Comment 4 The value of the data needs to be better highlighted throughout the manuscript by conducting the indicated statistical analysis. This is particularly needed when it comes to evaluating the LOD, the authors need to show whether the observed reading at the indicated LOD is of significant difference compared to the untreated control reading.

Comment 5 The authors need to better highlight (perhaps in a separate bar chart) the observed background/noise reading/luminescence at the chosen incubation period at different LPS concentrations. It is a bit confusing why the authors didn’t choose the incubation period of the highest S/N ratio for their assays?

Comment 6 While the authors highlighted in their methods a section for cryopreservation, they didn’t show any data for successful freezing/thawing of functional reporter cells. Such data can add valuable input, to further support the use of the chosen cancer cell line. For example, data for consistency and viability and functionality post thawing is needed.

Comment 7 In their method part the authors described two versions of virus production with nearly the same wording. At that point it is not clear whether and where two different viruses (lentivirus and undefined virus) were used. If not my mistake, it is strongly recommended to not use the exact wording for both descriptions.

Comment 8: Given the affiliation of the authors and the commercialization interest of Fuji Film in the space of stem cells, the authors should state clearly any potential conflict of interest and whether the work presented will/is planed to be commercialized.

6. PLOS authors have the option to publish the peer review history of their article (what does this mean? ). If published, this will include your full peer review and any attached files.

**Do you want your identity to be public for this peer review?** For information about this choice, including consent withdrawal, please see our Privacy Policy .

Reviewer #1: No

---

## [Author Response · Author response to Decision Letter 1]

30 Jan 2025

Our responses to the provided comments are listed in the "Response to Reviewer" section. Please review them.

---

## [Decision Letter · Decision Letter 1]

PONE-D-24-22098R1NOMO-1 combination with the Luciferase reporter gene offers a simple and rapid Monocyte Activation Test that can detect a wide range of pyrogen.PLOS ONE

Dear Dr. Nanao,

Thank you for submitting your manuscript to PLOS ONE. After careful consideration, we feel that it has merit but does not fully meet PLOS ONE’s publication criteria as it currently stands. Therefore, we invite you to submit a revised version of the manuscript that addresses the points raised during the review process.

We look forward to receiving your revised manuscript.

Kind regards,

Rui Tada, Ph.D.

Academic Editor

PLOS ONE

Reviewers' comments:

Reviewer's Responses to Questions

**Comments to the Author**

1. If the authors have adequately addressed your comments raised in a previous round of review and you feel that this manuscript is now acceptable for publication, you may indicate that here to bypass the “Comments to the Author” section, enter your conflict of interest statement in the “Confidential to Editor” section, and submit your "Accept" recommendation.

Reviewer #1: (No Response)

2. Is the manuscript technically sound, and do the data support the conclusions?

Reviewer #1: Partly

3. Has the statistical analysis been performed appropriately and rigorously? 

Reviewer #1: Yes

4. Have the authors made all data underlying the findings in their manuscript fully available?

Reviewer #1: Yes

5. Is the manuscript presented in an intelligible fashion and written in standard English?

Reviewer #1: No

6. Review Comments to the Author

Reviewer #1: The authors presented a revised version of their original draft in which they included important data comparing the expression levels of different TLRs between NOMO-1 cells and primary macrophages, which correspond to the current standard of MAT. In addition, they presented confirmatory data comparing the results of their reporter-based approach with standard ELISA measurements. Several questions raised during the review process were answered satisfactorily. However, their work as a whole still lacks sufficient depth and profound language revision.

It is important that the authors clarify whether their results are based on four technical or biological replicates. Biological replicates would include at least four different cell batches that were thawed and seeded, while technical replicates would mean that one frozen cell batch was distributed to four wells for simultaneous measurement. At this point, the present description does not allow an evaluation of whether the NOMO-1 cell line is a reliable and reproducible source for the MAT. Given the importance of the assay and the regulatory requirements it is of utmost importance to demonstrate reproducibility and standardization.

Overall, the manuscript requires a more thorough and critical discussion of its results, including an open acknowledgement of all limitations. In addition, a fundamental revision of the wording used in the text is strongly suggested. Amongst others, lines 23, 32, 175, 276, 290-295, 324-326, 334 and 429, were particularly difficult to comprehend or inadequately phrased. These are just exemplary lines! We therefore recommend revising the submitted work prior to publishing in the PLOS ONE journal.

Below, the authors may find additional helpful comments to improve the overall quality of their work:

Comment 1 – It should be noted that normalizing TLR or CD14 expression to β-actin levels is not an adequate approach to generally compare and draw conclusions about expression levels between different cell types, which may express β-actin to very different degrees. The authors could give information on the number of replicates for this assay to emphasize their conclusions.

Comment 2 – The authors referred to other established monocytic cell lines that included a reporter-based approach and stated that they performed a more thorough validation with more different TLR targets compared to them. However, a more functional comparison would have been informative, considering the availability of THP1 and MM6 cells in their first draft.

Comment 3 – The authors should include more up-to-date information regarding the European Pharmacopeia. They are no longer just committed to replacing RPT, but have officially announced in July 2024 that they will abolish the RPT.

Comment 4 – The authors display financial disclosures only for TO, although TN, YM, KS, TM and TN are also assigned to FUJIFILM Corporation.

Line 4 – pyrogens should be used as plural

Line 22 – pyrogens compromise endotoxins and non-endotoxins, as also later described

Line 23 – pyrogens do not necessarily enter immune cells to induce fever and endotoxic shock

32 – although it became clear later, “NF-kB reporter gene introduced cells” is hard to understand

345 – The detectability “of” NEPs

7. PLOS authors have the option to publish the peer review history of their article (what does this mean? ). If published, this will include your full peer review and any attached files.

**Do you want your identity to be public for this peer review?** For information about this choice, including consent withdrawal, please see our Privacy Policy .

Reviewer #1: No

---

## [Author Response · Author response to Decision Letter 2]

28 Mar 2025

In this manuscript, we have made extensive English revisions, including minor adjustments to the title.

---

## [Decision Letter · Decision Letter 2]

PONE-D-24-22098R2NOMO-1 cells expressing an NF-κB luciferase reporter gene facilitate a simple, rapid monocyte activation test that can detect a wide range of pyrogensPLOS ONE

Dear Dr. Nanao,

Thank you for submitting your manuscript to PLOS ONE. After careful consideration, we feel that it has merit but does not fully meet PLOS ONE’s publication criteria as it currently stands. Therefore, we invite you to submit a revised version of the manuscript that addresses the points raised during the review process.

We look forward to receiving your revised manuscript.

Kind regards,

Rui Tada, Ph.D.

Academic Editor

PLOS ONE

Journal Requirements:

Reviewers' comments:

Reviewer's Responses to Questions

**Comments to the Author**

1. If the authors have adequately addressed your comments raised in a previous round of review and you feel that this manuscript is now acceptable for publication, you may indicate that here to bypass the “Comments to the Author” section, enter your conflict of interest statement in the “Confidential to Editor” section, and submit your "Accept" recommendation.

Reviewer #1: (No Response)

2. Is the manuscript technically sound, and do the data support the conclusions?

Reviewer #1: (No Response)

3. Has the statistical analysis been performed appropriately and rigorously? 

Reviewer #1: (No Response)

4. Have the authors made all data underlying the findings in their manuscript fully available?

Reviewer #1: (No Response)

5. Is the manuscript presented in an intelligible fashion and written in standard English?

Reviewer #1: (No Response)

6. Review Comments to the Author

Reviewer #1: The authors have addressed most of the comments. I still ask to disclose any finical relationship for this scientific work.

It is not sufficient to disclose this in the cover letter nor I can see a financial disclose given to the PLOS one editorial office. The authors just say "yes", however a disclosure statement is requested and missing.

7. PLOS authors have the option to publish the peer review history of their article (what does this mean? ). If published, this will include your full peer review and any attached files.

**Do you want your identity to be public for this peer review?** For information about this choice, including consent withdrawal, please see our Privacy Policy .

Reviewer #1: No

---

## [Author Response · Author response to Decision Letter 3]

29 May 2025

In this review, there were no comments about the manuscript itself. Therefore, no changes have been made to the manuscript. Instead, we received a remark regarding the financial relationship for this scientific work. In response, we have requested the administrative office to update in the cover letter.

---

## [Editor Report · Decision Letter 3]

NOMO-1 cells expressing an NF-κB luciferase reporter gene facilitate a simple, rapid monocyte activation test that can detect a wide range of pyrogens

PONE-D-24-22098R3

Dear Dr. Nanao,

We’re pleased to inform you that your manuscript has been judged scientifically suitable for publication and will be formally accepted for publication once it meets all outstanding technical requirements.

Kind regards,

Rui Tada, Ph.D.

Academic Editor

PLOS ONE
---

## [Editor Report · Acceptance letter]

PONE-D-24-22098R3

PLOS ONE

Dear Dr. Nanao,

I'm pleased to inform you that your manuscript has been deemed suitable for publication in PLOS ONE. Congratulations! Your manuscript is now being handed over to our production team.

Kind regards,

on behalf of

Dr. Rui Tada

Academic Editor

PLOS ONE